# Infection Manager System (IMS) as a new hemocytometry-based bacteremia detection tool: A diagnostic accuracy study in a malaria-endemic area of Burkina Faso

**Annelies Post** [1,2☯]*, **Berenger Kaboré** [1,2,3☯], **Joel Bognini** [3], **Salou Diallo** [3], **Palpouguini Lompo** [3], **Basile Kam** [3], **Natacha Herssens** [4], **Fred van Opzeeland** [2,5], **Christa E. van der Gaast-de Jongh** [2,5], **Jeroen D. Langereis** [2,5], **Marien I. de Jonge** [2,5], **Janette Rahamat-Langendoen** [2,6], **Teun Bousema** [2,6], **Heiman Wertheim** [2,6], **Robert W. Sauerwein** [2,6], **Halidou Tinto** [3,7,8], **Jan Jacobs** [4,9], **Quirijn de Mast** [1,2], **Andre J. van der Ven** [1,2]*

1 Nijmegen Institute of International Health, Radboudumc, Nijmegen, the Netherlands, 2 Radboud Center for Infectious Diseases, Radboudumc, Nijmegen, the Netherlands, 3 IRSS/Clinical Research Unit of Nanoro (CRUN), Nanoro, Burkina Faso, 4 Department of Clinical Sciences, Institute of Tropical Medicine (ITM), Antwerp, Belgium, 5 Section of Pediatric Infectious Diseases, Laboratory of Medical Immunology, Radboud Institute for Molecular Life Sciences, Radboudumc, Nijmegen, the Netherlands, 6 Department of Medical Microbiology, Radboudumc, Nijmegen, the Netherlands, 7 Centre Muraz, Bobo-Dioulasso, Burkina Faso, 8 Institut Supérieur des Sciences de la Santé, Université Nazi Boni de Bobo-Dioulasso, Burkina Faso, 9 Department of Microbiology and Immunology, University of Leuven (KU Leuven), Leuven, Belgium

☯ These authors contributed equally to this work.
* annelies.post@gmail.com (AP); andre.vanderven@radboudumc.nl (AvV)

**Data Availability Statement:** All files are available from the DRYAD database (https://doi.org/10.5061/dryad.rjdfn2z9m).

## Abstract

### Background

New hemocytometric parameters can be used to differentiate causes of acute febrile illness (AFI). We evaluated a software algorithm–Infection Manager System (IMS)—which uses hemocytometric data generated by Sysmex hematology analyzers, for its accuracy to detect bacteremia in AFI patients with and without malaria in Burkina Faso. Secondary aims included comparing the accuracy of IMS with C-reactive protein (CRP) and procalcitonin (PCT).

### Methods

In a prospective observational study, patients of $\geq$ three-month-old (range 3 months– 90 years) presenting with AFI were enrolled. IMS, blood culture and malaria diagnostics were done upon inclusion and additional diagnostics on clinical indication. CRP, PCT, viral multi-plex PCR on nasopharyngeal swabs and bacterial- and malaria PCR were batch-tested retrospectively. Diagnostic classification was done retrospectively using all available data except IMS, CRP and PCT results.

### Findings

A diagnosis was affirmed in 549/914 (60.1%) patients and included malaria (n = 191) bacteremia (n = 69), viral infections (n = 145), and malaria-bacteremia co-infections (n = 47). The

**Funding:** AvV and QdM have a non-restricted research grant from SYSMEX corporation Europe, which was used to fund the current study. Sysmex contributed the prototype XN-450 and required reagents for the current study. The funding source was involved in the study design, but not in data collection, analysis, and interpretation of the data.

**Competing interests:** The authors have declared that no competing interests exist.

overall sensitivity, specificity, and negative predictive value (NPV) of IMS for detection of bacteremia in patients of $\geq$ 5 years were 97.0% (95% CI: 89.8–99.6), 68.2% (95% CI: 55.6–79.1) and 95.7% (95% CI: 85.5–99.5) respectively, compared to 93.9% (95% CI: 85.2–98.3), 39.4% (95% CI: 27.6–52.2), and 86.7% (95% CI: 69.3–96.2) for CRP at $\geq$20mg/L. The sensitivity, specificity and NPV of PCT at 0.5 ng/ml were lower at respectively 72.7% (95% CI: 60.4–83.0), 50.0% (95% CI: 37.4–62.6) and 64.7% (95% CI: 50.1–77.6) The diagnostic accuracy of IMS was lower among malaria cases and patients <5 years but remained equal to- or higher than the accuracy of CRP.

## Interpretation

IMS is a new diagnostic tool to differentiate causes of AFI. Its high NPV for bacteremia has the potential to improve antibiotic dispensing practices in healthcare facilities with hematology analyzers. Future studies are needed to evaluate whether IMS, combined with malaria diagnostics, may be used to rationalize antimicrobial prescription in malaria endemic areas.

## Trial registration

ClinicalTrials.gov (NCT02669823) https://clinicaltrials.gov/ct2/show/NCT02669823

## Author summary

This study describes the diagnostic accuracy of the Infection Manager System (IMS), a novel diagnostic algorithm for febrile illnesses that is equipped on a routine hematology analyzer. The latest generation hematology analyzers allow better differentiation between leukocyte subsets and their phenotype. The IMS was created, using differences in immune cell subsets (their activation status for instance), to differentiate viral from bacterial etiologies of fever. Such a tool may guide clinicians in their decision the initiate or withhold antimicrobial therapy.

The study was carried out among febrile patients aged 3 months and older in rural Burkina Faso, a sub-Saharan African setting where malaria is endemic. Standard microbiological techniques such as blood culture were used as a reference to assess the diagnostic accuracy of IMS. We then compared the diagnostic accuracy of the IMS with the marketed biomarkers C-reactive protein (CRP) and procalcitonin (PCT). Our study showed that the diagnostic performance of the IMS was similar to CRP and better than PCT to detect bacteremia in patients with and without malaria co-infection. Further studies are needed to see if the IMS can be safely used to guide initial antimicrobial treatment and help to reduce further spread of antimicrobial resistance.

## Introduction

Acute febrile illness (AFI) is an important health problem in sub-Saharan Africa (SSA). AFI can be caused by a variety of pathogens–bacteria, viruses, malaria parasites–which cause a non-specific clinical illness. Establishing the microbiological origin of AFI without laboratory diagnostics remains a challenge. While malaria remains common, there is an increasing appreciation for non-malarial causes of AFI in SSA [1,2] as well as for concurrent malaria and

bacteraemia [3–5]. The mortality of bacteremia is high and the prognosis depends on early recognition and treatment.

The introduction of malaria rapid diagnostic tests (RDTs) has greatly rationalized the use of anti-malarial drugs. However, other diagnostic tools for evaluation of AFI such as bacterial culture, are rarely available in SSA. Antibiotics are therefore regularly prescribed empirically among patients presenting with undifferentiated AFI. Even when malaria is suspected, antibiotics are commonly administered because of fear for concurring bacteremia [6]. The lack of microbiological tests indicating a bacterial infection is fueling antibiotic overuse and development of antimicrobial resistance (AMR) [7]. The global increase in AMR has been decreed an imminent threat to global health by the World Health Organization (WHO). As a result, development of rapid diagnostics for differentiation between bacterial- and non-bacterial AFI has become a priority [8].

An alternative to pathogen-specific (microbiological) diagnostics is to assess the host immune response to pathogens in peripheral blood. Biomarkers such as C-reactive protein (CRP) or procalcitonin (PCT) are advocated to guide antibiotic prescription, but their usefulness for patients with a concurrent malaria infection has been scarcely studied [9,10]. The host immune response can also be assessed by evaluating blood-cell morphology using hemocytometry. The latest generation Sysmex automated cell counters (hematology analyzers) are equipped with an enhanced panel of parameters detailing blood-cell differentiation, which was used to create a software algorithm–Infection Manager System (IMS)–to differentiate causes of AFI. The IMS has previously been tested in Indonesia, demonstrating its capacity to differentiate between arboviral and bacterial infection among adults[11]. Here we describe the diagnostic accuracy of the IMS to detect bacteremia and other bacterial infections in patients with AFI in a malaria endemic area in Burkina Faso, with reference to standard microbiological and clinical diagnostics, and compared to CRP and PCT.

## Methods

### Ethics statement

The study was performed in accordance with the declaration of Helsinki. The study was approved by the national ethical committee of Burkina Faso (ref 2015-01-006), the internal review board of IRSS (ref A03-2016/CEIRES) the ethical committee of the Antwerp University Hospital (ref 15/47/492) and the institutional review board of the Institute of Tropical Medicine Antwerp (ref 1029/15). Written informed consent was obtained from all participants or their parents/legal guardians prior to their inclusion.

### Study design

We performed a diagnostic accuracy study (clinicalTrials.gov, NCT02669823) at the Clinical Research Unit of Nanoro (CRUN) designed to assess the accuracy of two new Sysmex technologies: (1) a prototype hematology analyzer (XN-30) to directly detect malaria parasitized erythrocytes and (2) a marketed XN-450 hematology analyzer equipped with the IMS algorithm. This manuscript only includes the results obtained with the XN-450 hematology analyzer equipped with the IMS prototype. The performance of XN-30 has been published elsewhere [12].

The primary aim of the present study was to assess the diagnostic accuracy of the IMS for detecting bacteremia and malaria-bacteremia co-infection in a malaria endemic area among participants of five years and older. Secondary aims were to assess the diagnostic accuracy of the IMS for detection of (i) bacteremia among children < five years, (ii) bacterial infections,

and (iii) viral infections among both patients (i.e., below and ≥ 5 years old) as well as (iv) comparison of the diagnostic accuracy of the IMS to detect bacteremia with that of CRP and PCT.

## Study population and procedures

The Nanoro area in Burkina Faso [13] is hyperendemic for *Plasmodium falciparum* infections with peak incidences coinciding with the rainy season (July-October) [14]. Bacteremia among < five year old children is predominantly caused by non-Typhoidal *Salmonella* [15,16]. Participants were enrolled between March 2016 and June 2017 at the district hospital "Centre Medical avec Antenne Chirurgicale" (CMA) Saint Camille de Nanoro to which CRUN is affiliated [14]. Patients of three months and older with suspected AFI needing hospitalization were screened for eligibility. Patients were eligible if they had a measured temperature (auricular) of ≥38.0˚C or ≤35.5˚C, or a reported history of fever up to 48 hours prior to presentation, and suspicion of severe infection with signs of severe clinical illness including respiratory distress, prostration, altered consciousness, convulsions (one or more episodes), clinical jaundice, Systemic Inflammatory Response Symptoms (SIRS) criteria, severe malnutrition with severe anemia (hemoglobin <5 g/dl). Patients with fever lasting more than 7 days were excluded.

Upon inclusion, 2–5 ml EDTA anticoagulated blood was sampled for the index test, complete blood count, malaria diagnostics (thick- and thin blood films and RDTs) and blood culture. All samples were processed within one hour after sampling. A nasopharyngeal swab and aliquots of residual blood and plasma were stored at -80˚ for retrospective analyses. Additional diagnostics such as chest X-ray (CXR), abdominal echography, urinalysis, and culture of urine, stool, pus, or cerebrospinal fluid were performed on indication. Patients were followed daily during hospitalization and follow-up samples were taken if clinically indicated. Procedures for the reference tests used to diagnose the various underlying diseases are described in **S1 Text**.

Diagnostic classification was independently done by two study doctors (BK and AP) and an infectious disease specialist (QdM) after inclusion had been completed. In case of discordant results (10%), QdM assigned the decisive diagnosis. AP, BK and QdM used all available information except for CRP, PCT and IMS results. All cases were subjected to a "diagnostic classification", referring to the final diagnosis assigned by the researchers which therefore includes cases with uncertain etiology. The term "confirmed diagnosis" refers to any case in which the etiology of disease was confirmed through clinical signs (e.g., erysipelas), or microbiological/ radiological confirmation. The diagnostic classification scheme is described in **S1 Table**. Co-infections were defined as the presence of two or more confirmed infections. Newly diagnosed tuberculosis cases were considered bacterial infections. Newly diagnosed HIV infections were considered infection of unknown origin. Patients with HIV and a confirmed co-infection were classified according to the co-infection.

## Index test: Infection Manager System

The XN-series hematology analyzers (Sysmex Corporation, Kobe, Japan) can distinguish activated from non-activated cells by quantifying cellular activity and cell-membrane composition using fluorescence- and surfactant reagents that target RNA, DNA and bioactive lipid rafts [17]. This leads to further differentiation of cell lineages (see also **S2 Table**). The resulting enhanced panel of parameters detailing blood cell differentiation [17–21] was used to create an algorithm–the IMS—a software update that provides flags indicating presence or absence of an inflammatory response and subsequently classify the inflammation as matching malaria, bacterial or viral infection [18]. The output comprises a complete blood count (CBC) in combination with a flag and calculated likelihood score for viral and bacterial infections, see also **S2 Text**. The IMS reports 'Inflammation of unknown origin' when inflammation is flagged

but none of the likelihood scores match a decisive etiology. Data on performance of the malaria score are premature as the malaria score is still in early development. This manuscript reports the number of cases flagged as malaria or malaria co-infection but does not analyze its performance against malaria diagnostics.

The IMS algorithm was originally designed for adults [11]. To account for the rapid changes in blood composition during infancy, as well as differences in immunological response to pathogens between young infants and adults, the algorithm was converted to use absolute numbers of cell subsets rather than percentages. The main text reports the results of the converted algorithm, results of the original algorithm as tested in Indonesia are presented in **S3 Table**.

### Healthy control samples

A concurrent explorative cross-sectional field study to assess baseline hemocytometry data among a healthy population of one year and older was performed in the same study area (ClinicalTrials.gov Identifier: NCT03176719). Details on primary objectives will be reported elsewhere. A secondary objective was to assess the prevalence of subclinical malaria infections in the area. The blood samples were analyzed to assess how frequently IMS flagging inflammation was found among a healthy population with and without malaria parasitemia. Results were compared with plasma CRP levels.

### Statistical analysis

Data analysis was done according to a statistical plan agreed upon before data inspection. Data was analyzed using Stata 14 (Stata Corp, College Station, TX, USA). Differences in proportions and medians were compared using as appropriate a chi-square test, Mann-Whitney-U test, or student's t-test. Patients without a confirmed diagnosis were excluded from analysis. Sensitivity, specificity, positive predictive value (PPV) and negative predictive value (NPV) were assessed using the diagt-package (Stata). Cut-off values for CRP and PCT were defined prior to data inspection. Two cut-off values were used to predict a bacterial etiology of fever (1) 20 mg/L and 0.5 ng/ml plasma respectively as previously proposed in literature [22] and (2) the optimal cut-off value as determined by ROC analysis. Comparative analyses of diagnostic accuracy between the IMS and CRP/PCT were done using a McNemar test and reported as test-ratios with significance level. A significance level of 5% was used for all analyses.

### Results

A total of 930 patients (age ranged 3 months– 90 years old) were included between March 2016 and July 2017, sixteen of whom were subsequently excluded because of missing clinical data or IMS results, leaving 914 patients for analysis (**Fig 1**). Patients were subdivided into two age groups–< five years (n = 449) and ≥ five years (n = 465)–to account for age-related differences in blood composition and immune response. **Table 1** shows baseline characteristics and diagnostic classification for both age groups. The percentage of antimalarials taken in the past two weeks was higher among patients below five years (38.3% versus 28.8%; p = .002) whereas the number of patients who had taken antibiotics in the past two weeks was similar (39.6% vs 39.4%; p = .9).

In 343/449 (76.4%) patients below five years and in 206/465 (44.3%) patients of five years and older, a diagnosis was confirmed using pre-defined case definitions (**Table 1**). The pathogens causing bacteremia with (n = 47) and without (n = 69) concurrent malaria parasitemia are specified in **Table 2**. In total 191 patients had clinical malaria and an additional 12 malaria patients had a concurrent viral (n = 4) or non-bacteremic bacterial (n = 8) co-infection.

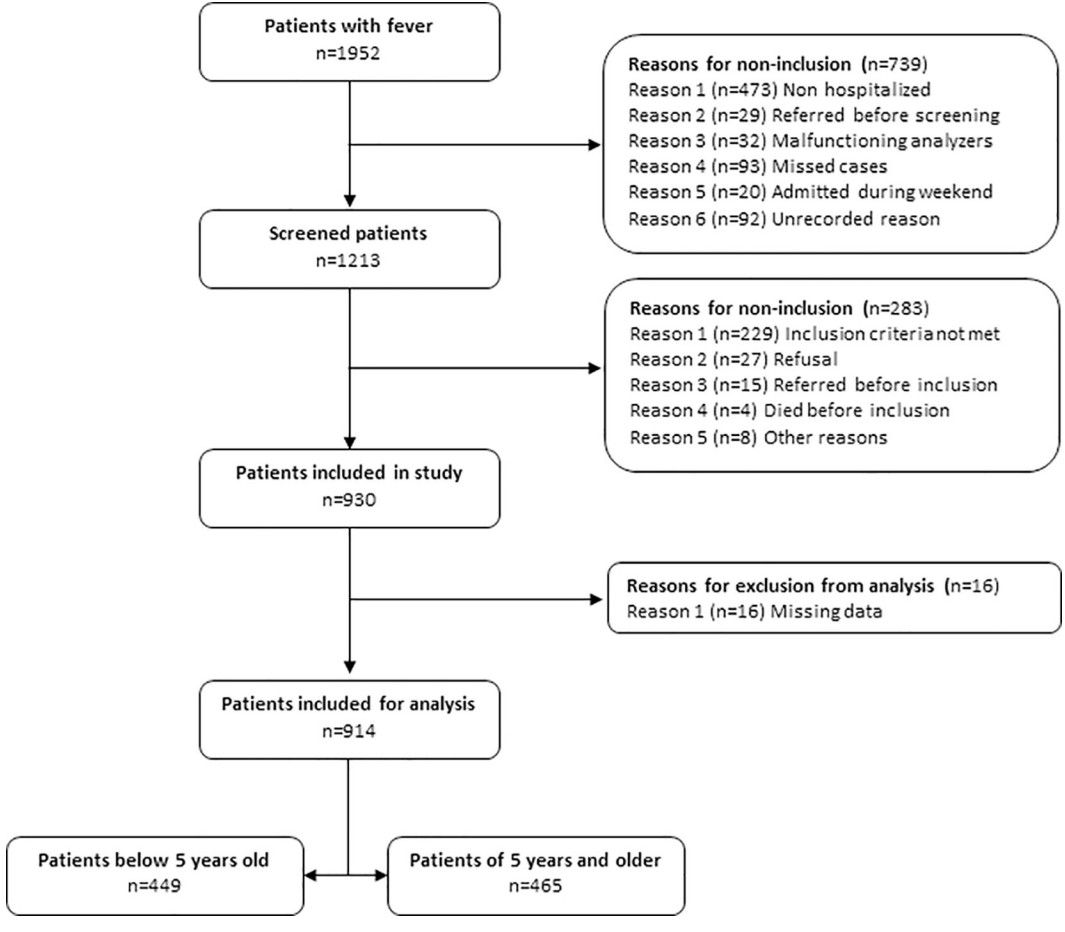

**Fig 1. Flow chart of inclusion.**

The IMS flagged inflammation matching bacterial infection in 318/914 (34.8%) (which includes both bacteremia and non-bacteremic bacterial infection), viral infection in 239/914 (26.1%) and malaria in 86/914 (9.4%) cases. In 50/914 (5.5%) patients the IMS found inflammation matching a combined malaria-bacterial infection. Inflammation of unknown origin was flagged in 221/914 (24.2%) patients consisting of 121 (54.8%) cases without- and 100 (45.2%) with a confirmed diagnosis (e.g., malaria (n = 46), viral infections (n = 27) bacterial infections (n = 22, including one bacteremia), and combined malaria-bacterial (n = 4) or malaria-viral (n = 1) infections).

Diagnostic accuracy of the IMS for detection of bacteremia with and without concurring malaria parasitemia.

**Fig 2** provides an overview of the diagnostic accuracy of the IMS to detect bacteremia by age group. The overall sensitivity of the IMS for detection of bacteremia was 92.2% (CI 95% 85.8–96.4) with a specificity of 70.9% (CI 95% 65.8–75.6), a PPV of 51.2% (CI 95% 44.2–58.2) and a NPV of 96.5% (CI 95% 93.5–98.4) among patients of all ages, with and without concurrent malaria parasitemia (**Table 3**). The sensitivity among ≥ five years old patients was higher than those < five years of age (97.1% [CI 95% 89.8–99.6] versus 85.4% [CI 95% 72.2–93.9]; p<0.0001), while the latter had a higher specificity (71.7% [CI 95% 66.1–76.9] versus 67.2% [CI 95% 54.6–78.2]; p < .0001). Concurrent malaria parasitemia decreased the accuracy to

**Table 1. Baseline characteristics and diagnostic classification by age group (n = 914).**

| Median (interquartile range (IQR) | Less than 5 years | 5 years and older |
|---|---|---|
| | n = 449 | n = 465 |
| Median age in months (<5 years) or years (≥5 years) | 18.8 (10.5–32.0) | 30 (13–52) |
| < 1 years old (n [%]) | 146 (32.5) | - |
| 1–5 years old (n [%]) | 303 (67.5) | - |
| 5–15 years old (n [%]) | - | 127 (27.3) |
| > 15 years old (n [%]) | - | 338 (72.7) |
| Male: Female ratio | 1.51 | 1.26 |
| Temperature at presentation (˚C) | 38.4 (38.0–39.4) | 38.3 (37.9–39.0) |
| Days of fever before presentation (days) | 2 (2–3) | 3 (2–4) |
| Antimalarials in past 2 weeks (n [%]) | 172 (38.3) | 134 (28.8) |
| Antibiotics in past 2 weeks (n [%]) | 178 (39.6) | 183 (39.4) |
| Systolic blood pressure (mm/Hg) | 98 (89–108) | 106 (96–120) |
| Diastolic blood pressure (mm/Hg) | 61 (55–67) | 66 (60–76) |
| Pulse (per min) | 124 (110–134) | 107 (98–122) |
| Respiratory rate (per min) | 34 (32–40) | 27 (26–29) |
| Malnutrition (n [%])* | 187 (41.6) | 133 (28.6) |
| Severe Acute Malnutrition (n [%]) | 90 (20.0) | 32 (6.9) |
| Current HIV at inclusion | 2 (0.5) | 14 (3.0) |
| Current Tuberculosis treatment | 0 | 6 (1.29) |
| **Diagnostic classification by age group** | | |
| **Biologically, clinically, or radiologically confirmed diagnoses** | **(n = 343/449, 76.4%)** | **(n = 206/465, 44.3%)** |
| **Malaria**[**] | **143 (41.7%)** | **48 (23.3%)** |
| **Bacteremia without malaria co-infection** | **20 (5.8%** | **49 (23.7%)** |
| **Bacteremia with malaria co-infection** | **28 (8.2%)** | **19 (9.2%)** |
| **Other bacterial infections** | **6 (1.7%)** | **66 (32.0%)** |
| Meningitis | 0 | 2 |
| Urinary tract infection | 0 | 6 |
| Gastroenteritis | 2 | 1 |
| Clinical diagnoses | 2 | 27 |
| Pneumonia | 2 | 14 |
| Tuberculosis | 0 | 16 |
| **Viral infection** | **138 (40.0%)** | **17 (8.3%)** |
| Respiratory tract infection[***] | 138 | 15 |
| Hepatitis | 0 | 2 |
| **Malaria-bacterial infection** | **3 (0.9%)** | **5 (2.4%)** |
| **Malaria-viral infection** | **2 (0.6%)** | **2 (0.9%)** |
| **Bacterial-viral infection** | **3 (0.9%)** | **0 (0.0%)** |
| **Uncertain etiology of disease** | **(n = 106/449, 23.6%)** | **(n = 259/465, 55.7%)** |
| **Infection of unknown origin** | **87 (82.1%)** | **143 (55.2%)** |
| Suspected bacterial infection | 10 | 61 |
| Suspected viral infection | 0 | 1 |
| Malaria with suspected other infection | 9 | 14 |
| Newly diagnosed HIV without bacteremia | 0 | 19 |
| Others | 68 | 48 |
| **Non-infectious diagnosis** | **4 (3.8%)** | **55 (21.2%)** |

(*Continued*)

**Table 1.** (Continued)

| Median (interquartile range (IQR) | Less than 5 years | 5 years and older |
|---|---|---|
|  | n = 449 | n = 465 |
| **Diagnosis unknown** | **15 (14.2%)** | **61 (23.6%)** |

\* For children < 5 years a Z-score weight/height of -2 or lower, for patients 5 years and older BMI z-score <-2was used. Severe Acute Malnutrition was defined as a Z-score weight/height of -3 or lower among children < 5 years or a BMI z-score of <-3

\*\* Reports only the cause of illness. In total 137 cases of qPCR confirmed submicroscopic malaria with parasite densities ranging from 0.007 to 13.7 were not included in this overview.

\*\*\* PCR on nasopharyngeal swab only performed for patients < 15 years old

detect bacteremia in both age groups; with a sensitivity of 90.0% (CI 95% 68.3–98.8) versus 82.1% (CI 95% 63.1–93.9) in patients < 5 years old (p < .0001) and 98.0% (CI 95% 89.1–99.9) versus 94.7% (CI 95% 74.0–99.9) among patients of 5 years and older (p < .0001).

There were a number of cases in which the IMS flagging for bacteremia was false negative and in which IMS flagging for bacterial infection was false positive ([Table 4]). Since the IMS cannot differentiate bacteremia from bacterial infections, a false-positivity rate for bacteremia alone cannot be calculated. In total 9/116 (7.7%) confirmed bacteremia cases were flagged as either viral infection (n = 6), malaria (n = 2) or infection of unknown origin (n = 1). The majority (n = 5) were invasive non-Typhoidal *Salmonella* (iNTS) cases among children below five years. Eight were blood culture confirmed, one was Polymerase Chain-Reaction (PCR) confirmed. In three cases antibiotics had been taken prior to inclusion to the study.

In total 103 non-bacterial cases were flagged as bacterial; they comprised patients with malaria (n = 64), viral respiratory tract infection (n = 37), and one patient each with viral hepatitis and combined malaria/viral infection.

**Table 2.** Overview of pathogens causing bacteremia in those with- and without concurrent malaria parasitemia (n = 116).

| | All bacteremia cases | | Bacteremia without concurrent parasitemia | | Bacteremia with concurrent parasitemia | |
|---|---|---|---|---|---|---|
| | Less than 5 years | 5 years and older | Less than 5 years | 5 years and older | Less than 5 years | 5 years and older |
| | n = 48 | n = 68 | n = 20 | n = 49 | n = 28 | n = 19 |
| *Salmonella* spp | 26[1] | 13[2] | 7 | 9 | 19 | 4 |
| *Escherichia coli* | 2 | 17 | 2 | 14 | 0 | 3 |
| *Neisseria meningitidis* | 2 | 1 | 1 | 1 | 1 | 0 |
| *Haemophilus influenzae* | 3[3] | 7[4] | 2 | 5 | 1 | 2 |
| *Staphylococcus aureus* | 0 | 7[5] | 0 | 5 | 0 | 2 |
| *Streptococcus pneumoniae* | 12[6] | 17[7] | 6 | 11 | 6 | 6 |
| Other Gram-negatives | 1 | 3 | 0 | 2 | 1 | 1 |
| Other Gram-positives | 1 | 2 | 1 | 2 | 0 | 0 |
| Mixed infection[8] | 1 | 1 | 1 | 0 | 0 | 1 |

1. 2/26 (7.7%) cases PCR confirmed | 2. 4/13 (30.8%) cases PCR confirmed | 3. 1/3 (33.3%) cases PCR confirmed | 4. 7/7 (100%) cases PCR confirmed | 5. 2/7 (28.6%) PCR confirmed | 6. 6/12 (50.0%) PCR confirmed | 7. 7/17 (41.2%) PCR confirmed. In total n = 29 (25.0%) were PCR confirmed, the rest (n = 87; 75.0%) were blood culture confirmed | 8. Mixed infections constituted blood cultures yielding more than one pathogen.

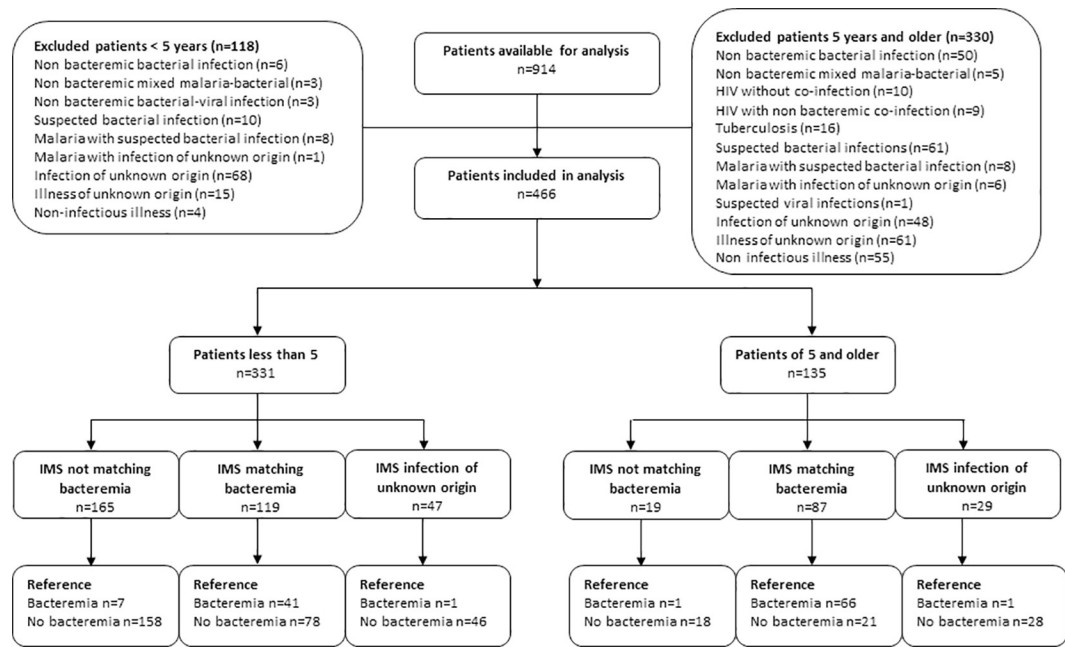

**Fig 2. Diagnostic accuracy of the IMS for bacteremia by age group.** <u>Legend:</u> IMS: Infection Manager System | "Reference" refers to blood culture confirmed cases.

## Comparison with CRP and PCT

CRP and PCT plasma levels were available for 883/914 (96.6%) patients. Among the different infections, median CRP levels of both bacteremia cases (107, IQR 44–142, n = 65) and all confirmed bacterial infections without malaria co-infection (97 mg/L, IQR 47–134; n = 109) were significantly higher than those obtained from clinical malaria cases (51 mg/L, IQR 22–100, n = 180; p < .0001 in both analyses). Samples of viral infections recorded the lowest CRP values (8 mg/L, IQR 3–23, n = 126).

Median PCT levels were highest among patients with malaria (3.5 ng/ml, IQR 0.6–18.2, n = 180) compared to patients with bacteremia- (1.9 mg/ml, IQR 0.5–8.5, n = 65), confirmed bacterial infections without malaria co-infection (0.9 ng/ml, IQR 0.4–3.9, n = 109), and viral infections (0.47 ng/ml, IQR 0.22–1.39). An overview of CRP and PCT values among the various types of infections can be found in **S1 Fig**.

The median (IQR) of CRP and PCT levels among patients with malaria, different causes of bacteremia (*Salmonella*, other Gram-negative and Gram-positive infections), and malaria-bacteremia co-infections is presented in **Fig 3A and 3B**. Malaria with bacterial co-infection produced slightly, though not significantly (p = 0.07) higher CRP plasma levels (72 mg/L, IQR 34–107, n = 43) compared to malaria alone (51 mg/L, IQR 22–100, n = 180), with considerable overlap between the two (**Fig 3A**). We found a wide spread of CRP levels between various causes of bacteremia (**Fig 3B**), with Gram-negative causes of bacteremia (e.g., *Salmonella*) having significantly lower CRP levels compared to Gram-positive causes of bacteremia (p = 0.02).

To compare the accuracy of the IMS to detect bacteremia with CRP and PCT, we performed a sub-analysis to the analysis performed in **Table 3**, using only patients with available CRP and PCT data (n = 445). This meant excluding a further 21 patients in whom insufficient plasma was available for CRP and PCT analysis (**Table 5**). At cut-off values of respectively ≥20 mg/L (CRP) and ≥0.5 ng/ml (PCT), the diagnostic accuracy of the IMS outperformed both

**Table 3. Diagnostic accuracy of the IMS for bacterial bloodstream infections presented by age group for patients with- and without malaria parasitemia (qPCR >0.05 p/uL).** The reference value is presented on the rows, the IMS result is presented in the columns.

| | IMS indicates bacterial | IMS indicates non-bacterial | IMS indicates bacterial | IMS indicates non-bacterial | IMS indicates bacterial | IMS indicates non-bacterial |
|---|---|---|---|---|---|---|
| Bacteremia among all cases (n = 466) | | | | | | |
| | Less than five years (n = 331) | | Five years and older (n = 135) | | All ages combined (n = 466) | |
| **Culture confirmed bacteremia** | 41 | 7 | 66 | 2 | 107 | 9 |
| **Non-bacterial infection** | 80 | 203 | 22 | 45 | 102 | 248 |
| Sensitivity (% [CI 95%]) | 85.4 (72.2–93.9) | | 97.1 (89.8–99.6) | | 92.2 (85.8–96.4) | |
| Specificity (% [CI 95%]) | 71.7 (66.1–76.9) | | 67.2 (54.6–78.2) | | 70.9 (65.8–75.6) | |
| PPV (% [CI 95%]) | 33.9 (25.5–43.0) | | 75.0 (64.6–83.6) | | 51.2 (44.2–58.2) | |
| NPV (% [CI 95%]) | 96.7 (93.3–98.6) | | 95.7 (85.5–99.5) | | 96.5 (93.5–98.4) | |
| AUC (CI 95%) | 0.79 ((0.73–0.84) | | 0.82 (0.76–0.88) | | 0.82 (0.78–0.85) | |
| Bacteremia among malaria negative cases (n = 198) | | | | | | |
| | Less than five years (n = 137) | | Five years and older (n = 61) | | All ages combined (n = 198) | |
| **Culture confirmed bacteremia** | 18 | 2 | 48 | 1 | 66 | 3 |
| **Non-bacterial infection** | 26 | 91 | 4 | 8 | 30 | 99 |
| Sensitivity (% [CI 95%]) | 90.0 (68.3–98.8) | | 98.0 (89.1–99.9) | | 95.7 (87.8–99.1) | |
| Specificity (% [CI 95%]) | 77.8 (69.2–84.9) | | 66.7 (34.9–90.1) | | 76.7 (68.5–83.7) | |
| PPV (% [CI 95%]) | 40.9 (26.3–56.8) | | 92.3 (81.5–97.9) | | 68.8 (58.5–77.8) | |
| NPV (% [CI 95%]) | 97.9 (92.4–99.7) | | 88.9 (51.8–99.7) | | 97.1 (91.6–99.4) | |
| AUC (CI 95%) | 0.84 (0.76–0.92) | | 0.82 (0.68–0.96) | | 0.86 (0.82–0.91) | |
| Bacteremia among malaria positive cases (n = 268) | | | | | | |
| | Less than five years (n = 194) | | Five years and older (n = 74) | | All ages combined (n = 268) | |
| **Culture confirmed bacteremia** | 23 | 5 | 18 | 1 | 41 | 6 |
| **Non-bacterial infection** | 54 | 112 | 18 | 37 | 72 | 149 |
| Sensitivity (% [CI 95%]) | 82.1 (63.1–93.9) | | 94.7 (74.0–99.9) | | 87.2 (74.3–95.2) | |
| Specificity (% [CI 95%]) | 67.5 (59.8–74.5) | | 67.3 (53.3–79.3) | | 67.4 (60.8–73.6) | |
| PPV (% [CI 95%]) | 29.9 (20.0–41.4) | | 50.0 (32.9–67.1) | | 36.3 (27.4–45.9) | |
| NPV (% [CI 95%]) | 95.7 (90.3–98.6) | | 97.4 (86.2–99.9) | | 96.1 (91.8–98.6) | |
| AUC (CI 95%) | 0.75 (0.67–0.83) | | 0.81 (0.73–0.89) | | 0.77 (0.72–0.83) | |

PPV: Positive Predictive Value | NPV: Negative Predictive Value | AUC: Area Under the Curve

CRP and PCT with an overall sensitivity of 91.8% (CI 95%: 85.0–96.2) versus 87.3% (CI 95%: 79.6–92.9) (p = .27) for CRP and 78.2% (CI 95%: 69.3–85.5) (p = .0026) for PCT, and a specificity of 71.6% (CI 95%: 66.4–76.4) versus 41.8% (CI 95%: 36.5–47.3) for CRP (p < .0001) and 36.1% (CI 95%: 31.0–41.5) for PCT (p < .0001). Just like for the IMS, the accuracy of CRP and PCT was lower in patients with malaria.

A sub-analysis using only patients of five years and older in whom both the IMS and CRP/PCT data was available (n = 131) is described in **Table 6**. The sensitivity of CRP in this sub-analysis was 93.9% (95%CI; 85.2–98.3) which was comparable to that of the IMS (97.0%; 89.5–99.6, p = .63), though the specificity of CRP was significantly lower (39.4% [27.6–52.2] compared to 68.2% [55.6–79.1]; p = .0005). The AUC of CRP was 0.67 (0.60–0.73), compared to 0.83 (0.77–0.89) for the IMS. The accuracy of PCT was lower with a sensitivity of 72.7% (60.4–83.0; p < .0001), a specificity of 50.0% (37.4–70.1; p = .029) and an AUC of 0.61 (0.53–0.70). Like in the IMS, the accuracy of CRP and PCT was lower in patients with malaria compared to

**Table 4. False negative flagging of IMS in cases with bacteremia (upper part) and false positive IMS flagging in cases of bacterial infection (lower part), among patients of all ages.**

**Bacterial pathogens detected by blood culture or PCR when the IMS bacterial inflammation flag was false negative.**

| Pathogen | (n)* | Age group | | Isolated using | | Coinciding malaria parasitemia | | Antibiotics prior to sampling | |
|---|---|---|---|---|---|---|---|---|---|
| | | <5 years | ≥5 years | Blood culture | PCR | Yes | No | Yes | No |
| Salmonella spp (all iNTS) | 5 | 5 | - | 5 | - | 2 | 3 | 1 | 4 |
| Escherichia coli | 2 | 1 | 1 | 2 | - | - | 2 | 2 | - |
| Haemophilus influenzae | 1 | - | 1 | - | 1 | 1 | - | - | 1 |
| Streptococcus pneumoniae | 1 | 1 | - | 1 | - | 1 | | - | 1 |

iNTS: invasive non-Typhoidal Salmonella | *n: Total number of false negative cases per bacterial species.

**False positive IMS bacterial inflammation flag presented against the classification according to reference standard.**

| Reference standard | n (%) * | IMS classification | | | |
|---|---|---|---|---|---|
| | | Bacterial | Malaria + bacterial | Bacterial + viral | Unknown |
| Malaria | 64/191 (33.5%) | 33 | 30 | - | 1 |
| Viral respiratory tract infection | 37/155 (23.9%) | 35 | 1 | - | 1 |
| Viral hepatitis | 1/2 (50.0%) | - | - | 1 | - |
| Malaria and viral infection | 1/4 (25.0%) | - | 1 | - | - |

*n (%): number of cases by etiology and percentage of total cases of that etiology

those without. In this sub-analysis, when compared with CRP, the IMS had similar sensitivity but higher specificity. Both CRP and IMS showed higher sensitivity and specificity than PCT. An exception was the sub-analysis among malaria negative patients where both CRP and PCT had one less false positive case compared to the IMS. The low number of cases in this sub-analysis (n = 58) led to a visually large (66.7% versus 75.0%) but non-significant (p = .56 and p = .65) difference in specificity.

Next, the accuracy of the IMS to detect all bacterial infections combined compared to CRP and PCT was calculated (**Table 7**). The sensitivity of the IMS was slightly lower than that of CRP but higher than PCT, and the specificity of the IMS was higher than both CRP or PCT. The AUC of the IMS was higher than that of both CRP and PCT. Missed cases by the IMS mainly consisted of patients with localized bacterial infections (n = 12), pneumonia (n = 8) and tuberculosis (n = 7).

The ROC curves of the IMS likelihood score for bacterial infection compared to CRP and PCT among patients of all ages without and with malaria parasitemia are presented in **Fig 4**. For patients without malaria parasitemia, IMS and CRP had comparable accuracy, whereas in patients with malaria parasitemia, IMS was more accurate than CRP. Of note, PCT was the least accurate in both settings. When using the ROC suggested cut-off values for CRP and PCT, the sensitivity, specificity, and AUC among patients >5 years old with and without malaria were 75.8% (95%CI 63.6–85.5), 72.7% (60.4–83.0) and 0.74 (0.67–0.82) for CRP and respectively 80.3% (68.7–89.1), 36.4% (24.9–49.1) with an AUC 0.58 (0.51–0.66) for PCT.

## Diagnostic accuracy of the IMS for viral infections

Finally, we compared the diagnostic accuracy of the IMS to detect viral infections to CRP and PCT at cut-off values of <20 mg/L and <0.5 ng/ml respectively. The sensitivity of the IMS was 51.3%, which was lower than CRP (64.6%, p = .013), and comparable to PCT (51.3%, p = 1.0). The specificity of the IMS (88.4%) was higher than both CRP (81.9%, p = .0071) and PCT

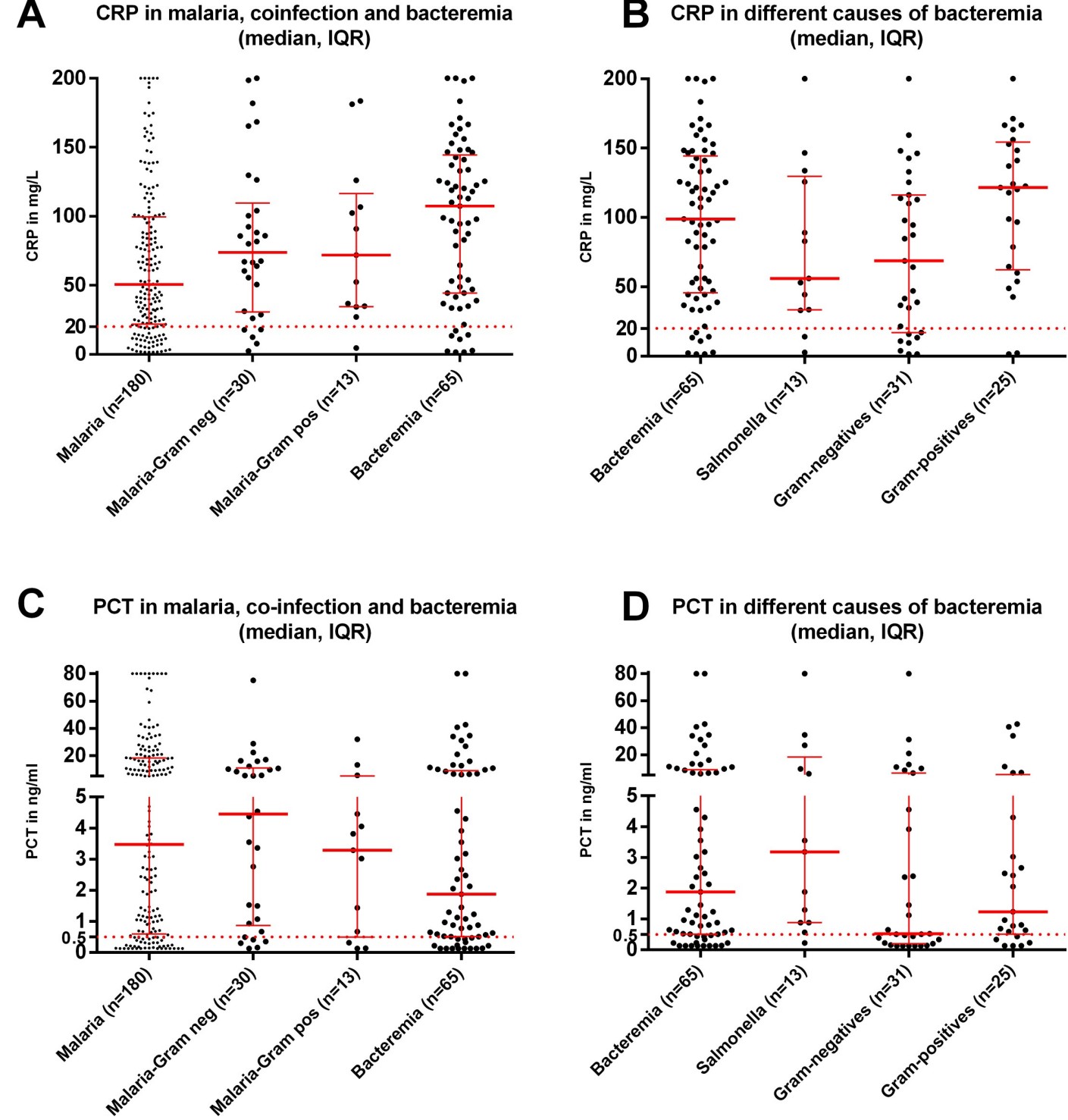

**Fig 3. Median (Interquartile range (IQR)) CRP levels among (A) malaria, co-infection, and bacteremia and (B) the different causes of bacteremia, and median (IQR) PCT (C) malaria, co-infection, and bacteremia and (D) the different causes of bacteremia.** <u>Legend</u>: the dotted line represents the cut-off values for CRP and PCT respectively.

**Table 5. Diagnostic accuracy of the IMS to detect bacteremia versus standard microbiological techniques among patients of all ages.** Data are presented separately for malaria positive cases, malaria negative cases and all cases, and compared to accuracy of CRP and PCT.

| | IMS indicating bacterial | | CRP | | test ratio | p value | PCT | | test ratio | p-value |
|---|---|---|---|---|---|---|---|---|---|---|
| **Bacteremia among all patients combined (n = 445)** | | | | | | | | | | |
| | **Positive** | **Negative** | **CRP >20** | **CRP<20** | **Ratio (95% CI)** | | **PCT>0.5** | **PCT<0.5** | **Ratio (95% CI)** | |
| **Confirmed bacteremia** | 101 | 9 | 96 | 14 | | | 86 | 24 | | |
| **Other infections** | 95 | 240 | 195 | 140 | | | 214 | 121 | | |
| Sensitivity (% [CI 95%]) | 91.8 (85.0–96.2) | | 87.3 (79.6–92.9) | | 0.95 (0.88–1.02) | .27 | 78.2 (69.3–85.5) | | 0.85 (0.77–0.94) | .0026 |
| Specificity (% [CI 95%]) | 71.6 (66.4–76.4) | | 41.8 (36.5–47.3) | | 0.58 (0.51–0.66) | < .0001 | 36.1 (31.0–41.5) | | 0.50 (0.43–0.59) | < .0001 |
| PPV (% [CI 95%]) | 51.5 (44.3–58.7) | | 33.0 (27.6–38.7) | | | | 28.7 (23.6–34.1) | | | |
| NPV (% [CI 95%]) | 96.4 (93.2–98.3) | | 90.9 (85.2–94.9) | | | | 83.4 (76.4–89.1) | | | |
| AUC (CI 95%) | 0.82 (0.78–0.85) | | 0.65 (0.60–0.69) | | | | 0.57 (0.52–0.62) | | | |
| **Bacteremia among malaria negative cases (qPCR <0.05 p/ul) (n = 190)** | | | | | | | | | | |
| | **Positive** | **Negative** | **CRP >20** | **CRP<20** | **Ratio (95% CI)** | | **PCT>0.5** | **PCT<0.5** | **Ratio (95% CI)** | |
| **Confirmed bacteremia** | 62 | 3 | 57 | 8 | | | 50 | 15 | | |
| **Other infections** | 28 | 97 | 39 | 86 | | | 59 | 66 | | |
| Sensitivity (% [CI 95%]) | 95.4 (87.1–99.0) | | 87.7 (77.2–94.5) | | 0.92 (0.85–0.99) | .025 | 76.9 (64.8–86.5) | | 0.80 (0.70–0.92) | .0018 |
| Specificity (% [CI 95%]) | 77.6 (69.3–84.6) | | 68.8 (59.9–76.8) | | 0.89 (0.77–1.02) | .1 | 52.8 (43.7–61.8) | | 0.68 (0.56–0.82) | .0001 |
| PPV (% [CI 95%]) | 68.9 (58.3–78.2) | | 59.4 (48.9–69.3) | | | | 45.9 (36.3–55.7) | | | |
| NPV (% [CI 95%]) | 97.0 (91.5–99.4) | | 91.5 (83.9–96.3) | | | | 81.5 (71.3–89.2) | | | |
| AUC (CI 95%) | 0.86 (0.82–0.91) | | 0.78 (0.73–0.84) | | | | 0.65 (0.58–0.72) | | | |
| **Bacteremia among malaria positive cases (qPCR >0.05 p/ul) (n = 255)** | | | | | | | | | | |
| | **Positive** | **Negative** | **CRP >20** | **CRP<20** | **Ratio (95% CI)** | | **PCT>0.5** | **PCT<0.5** | **Ratio (95% CI)** | |
| **Confirmed bacteremia** | 39 | 6 | 39 | 6 | | | 36 | 9 | | |
| **Other infections** | 67 | 143 | 156 | 54 | | | 155 | 55 | | |
| Sensitivity (% [CI 95%]) | 86.7 (73.2–94.9) | | 86.7 (73.2–94.9) | | 1.00 (0.87–1.15) | 1.0 | 80.0 (65.4–90.4) | | 0.92 (0.79–1.08) | 0.51 |
| Specificity (% [CI 95%]) | 68.1 (61.3–74.3) | | 25.7 (19.9–32.2) | | 0.38 (0.29–0.48) | < .0001 | 26.2 (20.4–32.7) | | 0.38 (0.31–0.49) | < .0001 |
| PPV (% [CI 95%]) | 36.8 (27.6–46.7) | | 20.0 (14.6–26.3) | | | | 18.8 (13.6–25.1) | | | |
| NPV (% [CI 95%]) | 96.0 (91.4–98.5) | | 90.0 (79.5–96.2) | | | | 85.9 (75.0–93.4) | | | |
| AUC (CI 95%) | 0.77 (0.71–0.83) | | 0.56 (0.50–0.62) | | | | 0.53 (0.46–0.60) | | | |

PPV: Positive Predictive Value | NPV: Negative Predictive Value | AUC: Area Under the Curve | CRP: C-reactive protein in mg/L | PCT: Procalcitonin in ng/ml.

(71.9%, p < .0001). There were too few cases of combined viral infection and malaria to perform a separate analysis.

In total 179/346 (51.7%) proven viral or malarial infections were correctly flagged as non-bacterial by the IMS, 142 (79.3%) of whom had been treated with antibiotics upon admission.

## IMS and CRP in a symptom free population

A total of 1003 healthy participants were included in the cross-sectional study. Malaria microscopy was performed on 927 of them; 483/927 (52.1%) had no microscopic parasitemia and 444/927 (47.9%) had asymptomatic parasitemia. The IMS flagged inflammation matching bacterial infection in 49/927 (5.3%) individuals, of whom 31 had a positive malaria microscopy. All others were flagged as 'no inflammation'. Sufficient plasma volume to measure CRP levels was available for 730 individuals: CRP levels of >20 mg/L were observed among 68/730 (9.3%) individuals, of whom 60 had positive malaria microscopy. These results suggest that the IMS has a lower false-positivity rate in a symptom free population compared to CRP.

**Table 6. Diagnostic accuracy of the IMS to detect bacteremia versus standard microbiological techniques among patients of 5 years and older.** Data are separately presented for malaria positive cases, malaria negative cases and all cases, and compared to accuracy of CRP and PCT.

| | IMS indicating bacterial | | CRP | | test ratio | p-value | PCT | | test ratio | p-value |
|---|---|---|---|---|---|---|---|---|---|---|
| **Bacteremia among patients of five years and older combined (n = 131)** | | | | | | | | | | |
| | Positive | Negative | CRP >20 | CRP<20 | Ratio (95% CI) | | PCT>0.5 | PCT<0.5 | Ratio (95% CI) | |
| **Confirmed bacteremia** | 64 | 2 | 62 | 4 | | | 48 | 18 | | |
| **Other infections** | 21 | 45 | 40 | 26 | | | 33 | 33 | | |
| Sensitivity (% [CI 95%]) | 97.0 (89.5–99.6) | | 93.9 (85.2–98.3) | | 0.97 (0.91–1.03) | .63 | 72.7 (60.4–83.0) | | 0.75 (0.65–0.86) | < .0001 |
| Specificity (% [CI 95%]) | 68.2 (55.6–79.1) | | 39.4 (27.6–52.2) | | 0.58 (0.42–0.79) | .0005 | 50.0 (37.4–62.6) | | 0.73 (0.57–0.95) | .029 |
| PPV (% [CI 95%]) | 75.3 (64.7–84.0) | | 60.8 (59.6–70.3) | | | | 59.3 (47.8–70.1) | | | |
| NPV (% [CI 95%]) | 95.7 (85.5–99.5) | | 86.7 (69.3–96.2) | | | | 64.7 (50.1–77.6) | | | |
| AUC (CI 95%) | 0.83 (0.77–0.89) | | 0.67 (0.60–0.73) | | | | 0.61 (0.53–0.70) | | | |
| **Bacteremia among malaria negative cases (qPCR <0.05 p/ul) (n = 58)** | | | | | | | | | | |
| | Positive | Negative | CRP >20 | CRP<20 | Ratio (95% CI) | | PCT>0.5 | PCT<0.5 | Ratio (95% CI) | |
| **Confirmed bacteremia** | 46 | 1 | 45 | 2 | | | 35 | 12 | | |
| **Other infections** | 4 | 8 | 3 | 9 | | | 3 | 9 | | |
| Sensitivity (% [CI 95%]) | 97.9 (88.7–99.9) | | 95.7 (85.5–99.5) | | 0.98 (0.94–1.02) | .32 | 74.5 (59.7–86.1) | | 0.76 (0.64–0.89) | .0009 |
| Specificity (% [CI 95%]) | 66.7 (34.9–90.1) | | 75.0 (42.8–94.5) | | 1.13 (0.75–1.68) | .56 | 75.0 (42.8–94.5) | | 1.13 (0.67–1.89) | .65 |
| PPV (% [CI 95%]) | 92.0 (80.9–97.8) | | 93.8 (82.8–98.7) | | | | 92.1 (78.6–98.3) | | | |
| NPV (% [CI 95%]) | 88.9 (51.8–99.7) | | 81.8 (48.2–97.7) | | | | 42.9 (21.8–66.0) | | | |
| AUC (CI 95%) | 0.82 (0.68–0.96) | | 0.85 (0.72–0.98) | | | | 0.75 (0.60–0.89) | | | |
| **Bacteremia among malaria positive cases (qPCR >0.05 p/ul) (n = 73)** | | | | | | | | | | |
| | Positive | Negative | CRP >20 | CRP<20 | Ratio (95% CI) | | PCT>0.5 | PCT<0.5 | Ratio (95% CI) | |
| **Confirmed bacteremia** | 18 | 1 | 17 | 2 | | | 13 | 6 | | |
| **Other infections** | 17 | 37 | 37 | 17 | | | 30 | 24 | | |
| Sensitivity (% [CI 95%]) | 94.7 (74.0–99.9) | | 89.5 (66.9–98.7) | | 0.94 (0.78–1.15) | .56 | 68.4 (43.4–87.4) | | 0.72 (0.54–0.96) | .025 |
| Specificity (% [CI 95%]) | 68.5 (54.4–80.5) | | 31.5 (19.5–45.6) | | 0.46 (0.31–0.68) | .0001 | 44.4 (30.9–58.6) | | 0.65 (0.48–0.88) | .0072 |
| PPV (% [CI 95%]) | 51.4 (34.0–68.6) | | 31.5 (19.5–45.6) | | | | 30.2 (17.2–46.1) | | | |
| NPV (% [CI 95%]) | 97.4 (86.2–99.9) | | 89.5 (66.9–98.7) | | | | 80.0 (61.4–92.3) | | | |
| AUC (CI 95%) | 0.82 (0.74–0.90) | | 0.60 (0.51–0.70) | | | | 0.56 (0.44–0.69) | | | |

PPV: Positive Predictive Value | NPV: Negative Predictive Value | AUC: Area Under the Curve | CRP: C-reactive protein in mg/L | PCT: Procalcitonin in ng/ml.

## Discussion

We performed a diagnostic accuracy study in Burkina Faso to assess the performance of a new diagnostic algorithm–the IMS–and the well-known biomarkers CRP and PCT to detect bacteremia among febrile ≥ five years old patients in a malaria endemic setting. We found that the IMS had a higher diagnostic accuracy to detect bacteremia than PCT at a cut of value of 0.5 ng/ml, and was comparable in sensitivity, but superior in specificity to CRP at a cut of value of 20 mg/L. Similar analysis in <5 years old patients as well as in those with concurrent malaria parasitemia showed a lower accuracy of both the IMS and CRP, though the accuracy of the IMS remained at least equal to- or higher than CRP for each sub-analysis. Combining the IMS and CRP did not significantly improve accuracy due to the high level of overlap between CRP and the IMS. The high NPV of IMS–also in non-bacteremic bacterial infections–suggests that the IMS holds promise to rationalize antimicrobial prescription in healthcare facilities where hematology analyzers are available. The relatively low specificity and PPV demonstrate that it is not (yet) suitable as a diagnostic for bacteremia.

**Table 7. Diagnostic accuracy of the IMS to detect all bacterial infections versus standard microbiological techniques among patients of all ages, presented for malaria positive cases, malaria negative cases and all cases, and compared to accuracy of CRP and PCT.**

| | IMS indicating bacterial | | CRP | | test ratio | p-value | PCT | | test ratio | p-value |
|---|---|---|---|---|---|---|---|---|---|---|
| **Bacterial infection among malaria positive and malaria negative cases combined (n = 528)** | | | | | | | | | | |
| | Positive | Negative | CRP>20 | CRP<20 | Ratio (95% CI) | | PCT>0.5 | PCT<0.5 | Ratio (95% CI) | |
| **Confirmed bacterial** | 148 | 45 | 164 | 29 | | | 129 | 64 | | |
| **Other infections** | 95 | 240 | 195 | 140 | | | 214 | 121 | | |
| Sensitivity (% [CI 95%]) | 76.7 (70.1–82.5) | | 85.0 (79.1–89.7) | | 1.11 (1.02–1.21) | .029 | 66.8 (59.7–73.4) | | 0.88 (0.79–0.97) | .018 |
| Specificity (% [CI 95%]) | 71.6 (66.5–76.4) | | 41.8 (36.5–47.3) | | 0.58 (0.51–0.66) | < .0001 | 36.1 (31.0–41.5) | | 0.50 (0.43–0.58) | < .0001 |
| PPV (% [CI 95%]) | 60.9 (54.5–67.1) | | 45.7 (40.4–51.0) | | | | 37.6 (32.5–43.0) | | | |
| NPV (% [CI 95%]) | 84.2 (79.4–88.2) | | 82.8 (76.3–88.2) | | | | 65.4 (58.1–72.2) | | | |
| AUC (CI 95%) | 0.74 (0.70–0.78) | | 0.65 (0.60–0.67) | | | | 0.51 (0.47–0.56) | | | |
| **Bacterial infection among malaria negative cases (qPCR <0.05 p/ul) (n = 252)** | | | | | | | | | | |
| | Positive | Negative | CRP>20 | CRP<20 | Ratio (95% CI) | | PCT>0.5 | PCT<0.5 | Ratio (95% CI) | |
| **Confirmed bacterial** | 97 | 30 | 108 | 19 | | | 80 | 47 | | |
| **Other infections** | 28 | 97 | 39 | 86 | | | 59 | 66 | | |
| Sensitivity (% [CI 95%]) | 76.4 (68.0–83.5) | | 85.0 (77.6–90.7) | | 1.13 (1.01–1.25) | .043 | 63.0 (54.0–71.4) | | 0.83 (0.73–0.95) | .014 |
| Specificity (% [CI 95%]) | 77.6 (69.3–84.6) | | 68.8 (59.9–76.8) | | 0.89 (0.77–1.02) | .12 | 52.8 (43.7–61.8) | | 0.68 (0.56–0.82) | .0001 |
| PPV (% [CI 95%]) | 77.6 (69.3–84.6) | | 73.5 (65.6–80.4) | | | | 57.6 (48.9–65.9) | | | |
| NPV (% [CI 95%]) | 76.4 (68.0–83.5) | | 81.9 (73.2–88.7) | | | | 58.4 (48.8–67.6) | | | |
| AUC (CI 95%) | 0.77 (0.72–0.82) | | 0.77 (0.72–0.82) | | | | 0.58 (0.52–0.64) | | | |
| **Bacterial infection among malaria positive cases (qPCR >0.05 p/ul) (n = 276)** | | | | | | | | | | |
| | Positive | Negative | CRP>20 | CRP<20 | Ratio (95% CI) | | PCT>0.5 | PCT<0.5 | Ratio (95% CI) | |
| **Confirmed bacterial** | 51 | 15 | 56 | 10 | | | 49 | 17 | | |
| **Other infections** | 67 | 143 | 156 | 54 | | | 155 | 55 | | |
| Sensitivity (% [CI 95%]) | 77.3 (65.3–86.7) | | 84.8 (73.9–92.5) | | 1.09 (0.94–1.28) | .33 | 74.2 (62.0–84.2) | | 0.96 (0.83–1.11) | .59 |
| Specificity (% [CI 95%]) | 68.1 (61.3–74.3) | | 25.7 (19.9–32.2) | | 0.38 (0.29–0.48) | < .0001 | 26.2 (20.4–32.7) | | 0.38 (0.31–0.49) | < .0001 |
| PPV (% [CI 95%]) | 43.2 (34.1–52.7) | | 26.4 (20.6–32.9) | | | | 24.0 (18.3–30.5) | | | |
| NPV (% [CI 95%]) | 90.5 (84.8–94.6) | | 84.4 (73.1–92.2) | | | | 76.4 (64.9–85.6) | | | |
| AUC (CI 95%) | 0.73 (0.67–0.79) | | 0.55 (0.50–0.61) | | | | 0.50 (0.44–0.56) | | | |

PPV: Positive Predictive Value | NPV: Negative Predictive Value | AUC: Area Under the Curve | CRP: C-reactive protein in mg/L | PCT: Procalcitonin in ng/ml.

Our primary outcome measure was bacteremia since it is difficult to diagnose, causes severe disease with a high mortality and requires early antibiotic treatment. Blood culture, the gold standard for detection of bacteremia, has a limited reliability as its sensitivity is only 40–60% [23–25]. Performing PCR on negative blood cultures increases the yield, but not up to 100% sensitivity. The high use of over-the-counter anti-microbials in the study area likely interferes with the results of blood culture and PCR. All these factors may lead to an underestimation of the number of bacteremia cases and may have affected our results. Furthermore, the high prevalence of asymptomatic malaria in our study area complicates reliability of our diagnostic classification process: some patients with asymptomatic malaria and undetected bacterial co-infection may have been falsely classified as clinical malaria.

The classification of viral respiratory infections was complex, as it is difficult to distinguish colonization from active infection using nasopharyngeal swabs. Viral respiratory tract infections are often complicated by bacterial superinfection which may have been unnoticed. We therefore cannot exclude that bacterial pneumonia cases were mistakenly classified as viral respiratory tract infections.

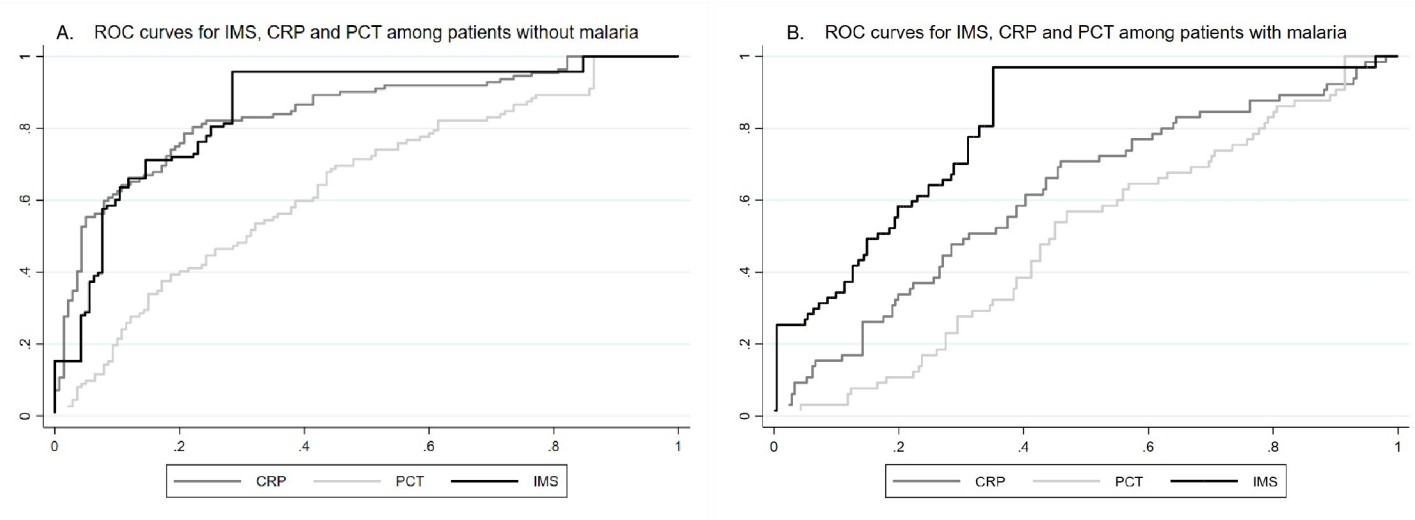

**Fig 4.** ROC curves for IMS, CRP and PCT among patients of all ages without (A) and with (B) malaria parasitemia.

Both early malaria and bacteremia lead to neutrophil mobilization and activation, which may explain the lower accuracy of the IMS in malaria patients. iNTS is often characterized by a limited innate immune response [23] which makes these intra-cellular infections more difficult to detect by the IMS as well as by CRP. This may explain the lower sensitivity among children below five years old as iNTS is the most prevalent cause of bacteremia among children below five years in the study area and is rare among (non-HIV infected) adults [15–16].

The Integrated Community Case Assessment (iCCM) program of WHO/UNICEF promotes the use of RDTs and early administration of antimicrobials for children [26]. This strategy has been effective in decreasing mortality but is now threatened by the increase in antimicrobial resistance (AMR). To counter the effects of AMR, the WHO recommends the development of Point of Care tests with a high NPV, to guide antibiotic prescription in patients with AFI [27]. The NPV of the IMS for detection of bacteremia ranges from 96.0–97.0% among the overall population, while the NPV for CRP is 90.0–91.5%. Both the IMS and CRP may therefore serve as tools to restrict antibiotics prescriptions. Both the IMS and CRP are influenced by presence of malaria parasitemia, though the large drop in specificity observed in CRP between patients with- and without malaria parasitemia suggests that the effect of parasitemia is stronger on CRP. Furthermore, our study also included healthy asymptomatic individuals of whom 9.3% had CRP levels of >20mg/L and 5.3% IMS flagging inflammation. Most participants with positive CRP or IMS result were malaria microscopy positive, supporting the confounding effect of malaria on CRP or IMS results. A combined malaria/CRP rapid diagnostic test would be a breakthrough for rural settings, though the determination of a valid cut-off value for CRP will be challenging.

The IMS is a learning algorithm which has the potential to increase its accuracy as more data is fed into the algorithm. Further development of the IMS will be directed to increase the accuracy of detecting combined bacterial infections in patients with bacteremia. Additionally, the impact of other factors which may influence blood counts such as (hematological) malignancies, chronic illnesses such as diabetes, cardiovascular diseases, and the use of immunomodulatory drugs on the accuracy of the IMS still need to be assessed.

Hematology analyzers are presently routine equipment of laboratories in peripheral hospitals up to national referral laboratories where second and third-line antimicrobials are most frequently prescribed. CBCs are commonly requested in these setting in patients with AFI and

simultaneous reporting of the IMS could further support health workers in decision making at similar costs as for CBC. The fact that hematology analyzers are amongst the most frequently used diagnostic equipment in sub-Saharan Africa means that the infrastructure required to operate the IMS is already in place, making it relatively easy to implement. Furthermore, the new hematology analyzers can be linked to internet and provide data to central governmental epidemiological and diseases control units.

## Limitations

In addition to the limitations to the study environment mentioned in the second Alinea there were several other limitations to the study design.

First, the final diagnoses were made retrospectively which may have decreased their reliability. Second, the limited diagnostic possibilities in the study area decreased the number of definitive diagnosis. Additionally, the number of proven viral infections may be lower than expected which may limit the reliability of the analysis for viral infections. Furthermore, this study did not explore the possible immunomodulatory effect of antimicrobials taken prior to inclusion and its potential influence on the IMS results.

Finally, while the sub-analyses performed for age and malaria demonstrate their influence on the accuracy of all three tested techniques, they also decrease the numbers of available cases per analysis thereby decreasing the reliability of these sub-analyses.

## Conclusion

The IMS algorithm is a new diagnostic tool to differentiate viral and bacterial infections in malaria endemic areas. Its sensitivity is similar to CRP using 20 mg/L as cut-off value, but it is considerably better at excluding bacteremia due to its high NPV and specificity. Future studies should evaluate whether CRP and IMS can be used to rationalize antibiotics use.

## Supporting information

**S1 Text. Sample collection, case definitions and diagnostic procedures.**
(DOCX)

**S2 Text. Technical supplement on the development of the IMS.**
(DOCX)

**S1 Table. Diagnostic classification schema with criteria for the different infections.**
(DOCX)

**S2 Table. Overview of novel parameters determined by the XN haematology analyser series.**
(DOCX)

**S3 Table. Results original IMS algorithm based on absolute numbers: Diagnostic accuracy for bacterial bloodstream infection including bacteraemia/malaria co-infections, bacterial infections and viral infections.**
(DOCX)

**S1 Fig. Median (IQR) CRP (A) and PCT (B) values among clinical malaria, bacterial infections, bacteremia, and viral infections. Legend: the dotted line represents the cut-off values for CRP and PCT respectively.**
(TIF)

## Acknowledgments

The authors would like to thank all study participants for their participation. We furthermore thank the nurses from Centre Medicale avec Antenne Chirurgicale de Nanoro and the laboratory technicians, team of data managers and the study nurses from the Clinical Research Unit of Nanoro–Bakombania Abassiri, Esther Kapioko, Celine Nare, Catherine Nikiema and Clement Zongo for their dedication to the study. Additionally, we thank Mike Berendsen, Helga Dijkstra, Heidi Lemmers and Kjerstin Lanke from Radboudumc for technical support.

## Author Contributions

**Conceptualization:** Annelies Post, Berenger Kaboré, Palpouguini Lompo, Basile Kam, Natacha Herssens, Jan Jacobs, Quirijn de Mast, Andre J. van der Ven.

**Data curation:** Berenger Kaboré, Joel Bognini, Salou Diallo, Palpouguini Lompo, Basile Kam, Fred van Opzeeland, Christa E. van der Gaast-de Jongh.

**Formal analysis:** Annelies Post, Berenger Kaboré, Basile Kam, Quirijn de Mast, Andre J. van der Ven.

**Funding acquisition:** Jan Jacobs, Quirijn de Mast, Andre J. van der Ven.

**Supervision:** Halidou Tinto, Jan Jacobs, Quirijn de Mast, Andre J. van der Ven.

**Validation:** Jeroen D. Langereis, Marien I. de Jonge, Janette Rahamat-Langendoen, Teun Bousema, Heiman Wertheim, Robert W. Sauerwein.

**Visualization:** Annelies Post, Berenger Kaboré, Basile Kam, Andre J. van der Ven.

**Writing – original draft:** Annelies Post, Berenger Kaboré, Basile Kam, Andre J. van der Ven.

**Writing – review & editing:** Annelies Post, Berenger Kaboré, Joel Bognini, Salou Diallo, Palpouguini Lompo, Basile Kam.

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
