## [Decision Letter · Decision Letter 0]

9 Nov 2020

Dear Post,

Thank you very much for submitting your manuscript "A new hemocytometry based tool to detect or exclude bacteremia in children and adults with and without malaria; a diagnostic accuracy study from Burkina Faso." for consideration at PLOS Neglected Tropical Diseases. As with all papers reviewed by the journal, your manuscript was reviewed by members of the editorial board and by several independent reviewers. The reviewers appreciated the attention to an important topic. Based on the reviews, we are likely to accept this manuscript for publication, providing that you modify the manuscript according to the review recommendations. 

Sincerely,

Jeanne Salje

Deputy Editor

Jeanne Salje

Deputy Editor

Reviewer's Responses to Questions

**Key Review Criteria Required for Acceptance?**

**Methods**

-Are the objectives of the study clearly articulated with a clear testable hypothesis stated?

-Is the study design appropriate to address the stated objectives?

-Is the population clearly described and appropriate for the hypothesis being tested?

-Is the sample size sufficient to ensure adequate power to address the hypothesis being tested?

-Were correct statistical analysis used to support conclusions?

-Are there concerns about ethical or regulatory requirements being met?

Reviewer #1: The objectives were clearly stated.

The study design was appropriate.

Sample size calculation was not provided. Small number of subjects for some comparisons may affect the study power, however, this was acknowledged as a limitation.

Specific comments on methods are provided below (please refer to Summary and General Comments).

Reviewer #2: Methods provided are good. Although citations have been given the described methods , it is important to briefly describe the steps involved, and provide information regarding the brand and catalogue number of materials and equipment used in the study (e.g. dyes, the brand of the device for CBC, CRP, PCT, microscopy etc). A table listing these materials/ equipment with their sources & catalogue number is useful for readers that want to replicate the study.

Reviewer #3: The study design is appropriate to the stated objectives.

**Results**

-Does the analysis presented match the analysis plan?

-Are the results clearly and completely presented?

-Are the figures (Tables, Images) of sufficient quality for clarity?

Reviewer #1: Specific comments are provided below (please refer to Summary and General Comments).

Reviewer #2: The analyses matched the analysis plan. The data presentation could be improved and additional graphs are needed to convey the message more clearly, which should not be difficult to do. Additional requests as follows:

Page 14: “comparison with CRP and PCT”:

The statement “Median CRP levels were the highest in bacterial infections……”. This is a little bit confusing after referring to Figure 3, as it seems like the higher levels were attributed to the Gram-positive bacteria and not the Gram-negative bacteria. Authors should discuss about this as well. 

Besides, which bacteria is the median value provided in the text referring to? Or the value stated is based on analyses of all bacterial infections as a whole? If this were the case, the graph should be presented accordingly, or a separate plot is provided as a supplement. 

It is better to present the graph in scatter plots so that readers can see the distribution of the data points more clearly. I strongly recommend the authors to change the graph presentation to scatter plot. 

Also, in the text, the CRP value of viral infections was mentioned but it was not shown in Figure 3. This should be included, along with modifications on the figure legend. 

There should be another similar graph describing PCT of samples tested for easier and clearer comparison.

Reviewer #3: The analysis matches the analysis plan.

**Conclusions**

-Are the conclusions supported by the data presented?

-Are the limitations of analysis clearly described?

-Do the authors discuss how these data can be helpful to advance our understanding of the topic under study?

-Is public health relevance addressed?

Reviewer #1: The conclusions were supported by data and results.

The limitations were acknowledged.

Specific comments are provided below (please refer to Summary and General Comments).

Reviewer #2: conclusions drawn are clear, and limitations are discussed. For discussion:

Other concerns/ issues that may affect the accuracy of this cellular immune response-based detection tool can be discussed more, such as application of this method on immuno-compromised groups (genetic and pathogen-induced), people with underlying conditions such as allergy, cardiovascular problems, diabetes mellitus, whose blood cellular components’ immuno-activity profiles may be different from those of normal, healthy individuals. These may be important points for medical workers from other countries to consider if they want to try this technique. 

For individual harboring malaria parasites (and also all other individuals recruited actually), do the authors know if the subjects had taken any anti-malarial (or any other drugs) prior to blood collection? Some anti-malarials and drugs have immuno-modulatory properties, which may confound the results collected. 

"Age" seems to be an important factor to consider when using this method. What are the authors' opinions regarding application of this method in places with large elderly population? 

It would be good to discuss briefly possible ways to improve this detection tool.

Reviewer #3: The conclusions are supported by the data presented.

**Editorial and Data Presentation Modifications?**

Reviewer #1: Specific comments are provided below (please refer to Summary and General Comments).

Reviewer #2: It would be easier for the review process if line number and page numbers are provided in the manuscript.

Title page: The title does not quite highlight the core focus of this manuscript, which is the IMS. Here’s my suggestion for the authors' consideration:

“Infection Manager System (IMS) as a new hemocytometry-based bacteremia detection tool in clinical setting: a diagnostic accuracy study in a malaria-endemic area of Burkina Faso”. 

Abstract: the clinical trial reference number in the abstract can be removed.

Author summary: please standardize the usage of abbreviation: “procalcitonin” should be “PCT” after the first mentioning of "procalcitonin".

Introduction:

2nd sentence: “AFI can be caused by a variety of pathogens- bacterial, viral, malaria-…”; “malaria parasites” or “Plasmodium spp.” may be more suitable to be addressed as “pathogens”. 

Last sentence of first paragraph: “the mortality of bacteremia is high and outcome depends on early recognition and treatment”, I suggest addition of “prognosis” before "outcome". 

2nd paragraph, 3rd sentence: “Even when severe malaria is diagnosed, antibiotics are commonly administered because of fear for co-occurring bactermia”. The word “co-occurring” used throughout the manuscript may be replaced with “concurring”. Can you provide a reference for this particular statement?

3rd paragraph, 4th sentence: the sentence should be started with a “The”.

Methods:

Please rephrase the third sentence of this segment to make it clearer.

Results:

Table 4 (page 15), Table 5 (page 17) and Table 6 (page 19): 

Clear descriptions should be given regarding what the values within each bracket is representing. 

Please standardize the usage of abbreviation throughout the text. “procalcitonin” should be “PCT”. 

Page 16:

The 2nd sentence, typing mistake: “The sensitivity of CPR…..” should be “CRP” instead.

The 6th sentence: the message may be clearer if it is split into two shorter sentences: In this sub-analysis, when compared with CRP, IMS had similar sensitivity but higher specificity. Both CRP and IMS showed higher sensitivity and specificity than PCT...”

Page 20:

2nd sentence: “When using the ROC…..patients >5…”, “kindly add “years old” after the word “5”. 

For section "IMS and CRP in a healthy control population":

The subheading is misleading. The samples aren’t really “healthy controls” as some of them are infected with parasites, albeit without showing signs and symptoms. I suggest changing it to “symptom-free population”. 

In this segment as well, please give a brief concluding remark on the comparisons of different methods used.

Discussion:

Please use “cut-off” value throughout your manuscript when referring to “cut-off values”. 

Page 21:

The short paragraph, “Both early malaria…..in malaria patients”, followed by “iNTS is…..as well as by CRP”. A "bridge" is needed between these two sentences (first sentence referred to malaria, second sentence referred to INTS) to make the flow of the discussion content better. 

Page 22, last sentence: “Both IMS and CRP…..”: the word “(a)symptomatic” can be removed.

Also, in the same sentence, regarding the effect of parasitemia being stronger on CRP, is it backed by statistical analysis? If it is, please state it clearly in the text. 

In the same paragraph, this sentence may be better if the word “confounding” is added:

“Most participants with positive CRP or IMS results were malaria microscopy positive, supporting the confounding effect of malaria on CRP and IMS results”. 

Others:

In supplementary data, page 4: a full description for RE-MONO/ RE-Mono should be given before using abbreviation, and the abbreviation used should be in consistent font format.

Reviewer #3: Extensive English revision is needed.

**Summary and General Comments**

Reviewer #1: General:

This manuscript describes the evaluation of a created software algorithm, called Infection Manager System (IMS) to detect bacteremia in acute febrile illness patients with and without malaria in Burkina Faso. The results were evaluated against CRP and PCT. This study is a part of a diagnostic accuracy study. The authors have previously validated the developed IMS (against CRP & PCT) to differentiate between arboviral and bacterial infections among Indonesian patients aged 14 years and above (Prodjosoewojo et al. PLoS NTDs. 2019), and also evaluated XN-30 hematology analyzer for malaria detection (Post et al. BMC medicine. 2019). More studies will also be sent for publication. The study concluded that IMS had similar sensitivity and better specificity when compared to CRP; however, the authors stated that it is not (yet) suitable as a diagnostic for bacteremia.

Overall, the study is interesting and important. The manuscript is well-written, although the tables and classification of groups and comparisons can be confusing. Specific comments are provided below.

SPECIFIC COMMENTS:

1. ABSTRACT:

1.1. patients of ≥ three-month-old (age range can be used here). 

1.2. PCT performed significantly worse. This statement can be replaced by actual results. Does this mean that IMS & CRP results were comparable?!

1.3. “The diagnostic accuracy was lower among malaria cases ….” Be specific and provide complete sentences. diagnostic accuracy of IMS.

2. Author summary:

2.1. “Complete blood count is globally the most common requested laboratory examination.” Can be either removed or explained. Maybe it is related to statements mentioned in the last paragraph of discussion! to support the integration of hematology analyzer with the IMS algorithm.

2.2. “to reduce antimicrobial resistance formation” this can be rephrased. 

3. METHODS:

3.1. Study design: Why the diagnostic accuracy of IMS among ≥ 5 years was considered the primary objective while in < 5 years was a secondary objective? The results involved both groups!! and the number of patients aged < 5 (331) was higher than those aged ≥ 5 years (135)!! Was this due to lower IMS accuracy reported among < 5 patients? but the design was decided before the commencement of data collection!!

3.2. Study population and procedures: It is stated that “Patients were eligible if they ……… severe malnutrition with severe anemia.” Please revise this statement! In line with this statement, malnutrition prevalence presented in table 1 (41.6% in < 5 and 28.6% in ≥ 5 years) indicates severe malnutrition. And according to footnote, Z-score weight/height was used; i.e. the malnutrition here was the severe acute malnutrition (SAM). The reported proportions (41.6% & 28.6%) seem to be very high.!! Please revise.

3.3. Study design: “among both patients below and over five years old as well as ….”. to include those aged 5 years, this can be changed to “among both patients (i.e. below and ≥ 5 years old) as well as ….”

3.4. Study population and procedures: “malaria diagnostics”! These were mentioned in the supplementary file, but methods names can be mentioned between brackets. 

4. RESULTS:

4.1. First paragraph and table 1: p values are stated in the text but table 1 displays no values. A column can be added to Table 1 to present p values.

4.2. In the text and tables 4-6, report all p-values to three decimals. 

4.3. In the text, “…….. (n=55) led to a visually large (66.7% versus 75.0%) …but n= 58 in table 5!

4.4. What was the patients’ age range (youngest & oldest)?

4.5. Do you expect a difference between patients aged 5-15 years (n = 127) and those aged > 15 years (n = 338)? 

4.6. The title stated “children” and “adult” but the results somehow did not clearly reflect such age grouping. The comparison was between < 5 and ≥ 5 years, with adults were included in the second group!!

4.7. Table 1A: “Median (IQR 25-75)” !! whats this?

4.8. Text & table 2: “The sensitivity among � five years old patients was higher than those < five years of age, while the latter patients had a higher specificity.” Why statistical tests were not used to compare the differences between both patients groups? 

4.9. Table 2: Bacteremia among malaria positive cases (n=268), with Less than five years (n=194) and Five years and older (n=74). How these 194 and 74 malaria positives can be extracted from table 1?

4.10. Add “n” of each of the 3 analysis groups, i.e. Bacterial infection among all cases; Bacterial infection among malaria negative cases and Bacterial infection among malaria positive cases. I have some confusion with Tables 4 and 6!!

5. DISCUSSION:

5.1. First paragraph: “The relatively low specificity and PPV demonstrate that it is not (yet) suitable as a diagnostic for bacteremia.”

5.2. Second paragraph: some limitations of the study findings are well-described in this paragraph but not mentioned in the limitation section!!

5.3. Third paragraph: “Both early malaria and bacteremia both …” remove the repeated “both”.

5.4. Third paragraph: “….. some patients with asymptomatic malaria and undetected bacterial co-infection may have been falsely classified as malaria.”! Do you mean as malaria mono-infection? please rephrase to improve clarity.

5.5. Fourth paragraph: “… (a)symptomatic …” !!

Reviewer #2: This manuscript described a software algorithm that can rapidly detect and differentiate between bacterial and non-bacterial acute febrile illness based on the phenotypic characteristics of cellular components in the blood. This cost-effective method is potentially useful in clinical settings as a quick reference for medical workers in deciding treatment of choice for the patients, hence reducing the tendency of over-using antibiotics that can lead to selection of drug-resistant bacteria in the community. Nevertheless, there are issues and concerns that may confound the accuracy and usefulness of this method in a broader clinical setting, which need to be addressed more carefully. 

As cited in the manuscript, this method was previously used to differentiate between arboviral and bacterial febrile illness in Indonesia, detecting malaria in a well-controlled human experiment setting, as well as distinguishing between bacterial and viral infections in pediatric setting. In this manuscript, the authors evaluated the efficiency of this method to differentiate between bacterial and non-bacterial acute febrile illness in a setting endemic to malaria, which could confound the method. Besides, the authors also performed a patient age-group based comparison to evaluate the effect of age on the accuracy of the method. 

Overall, this manuscript provides important information to the relevant fields, which has potential to improve the quality of patient management in hospitals, provided that the platform is perfected, which seems to be feasible in a near future. 

A few suggestions were given for the improvement of this manuscript. With some minor revisions and modifications, the manuscript may be suitable for publication.

Reviewer #3: This manuscript describes the result of a clinical study to evaluate the diagnostic accuracy of the Infection Manager System (IMS), a novel diagnostic algorithm for febrile illnesses that is equipped on a routine hematology analyzer. The study was conducted with AFI patients with and without malaria in Burkina Faso. Indeed, the development of an easy, accurate and affordable diagnostic system is very important. From this ms, the data shows that IMS is reasonably accurate with higher specificity than CRP diagnosis. However, the manuscript is very premature and needs extensive improvement/revision.

Major criticisms

1. Throughout the whole manuscript, there are so many typographical and grammatical errors with confusing syntax. In addition, there are no page numbers and line numbers. Examples picked up in first 3 pages are below: 

a. PCT performed significantly worse.

b. The IMS was tested against standard microbiological techniques.

c. Complete blood count is globally the most common requested laboratory examination.

d. The trend of increasing AMR has been decreed an imminent threat to global health Here we describe the diagnostic accuracy of IMS to detect bacteremia and bacterial infections in general in patients with AFI in a malaria endemic area in Burkina Faso,

2. There are no description for figure legends.

3. Many abbreviations are not defined without full names.

4. Rational for “Newly diagnosed HIV infection was considered infection of unknown origin” should be described/further elaborated. Why?

5. The first line of Table 1; Median age (months/years) is very confusing. Less than 5 years ;18.8. Is this month? 5 years and older; 30. Is this year?

6. The discussion can also be used by the authors to provide their thoughts/interpretation on the possible improvement of XN-30 equipped with IMS algorithm.

7. Figure 4 describes ROC curves of the IMS likelihood score for bacterial infection compared to CRP and PCT for all groups. Authors should show different ROCs with or without malaria.

PLOS authors have the option to publish the peer review history of their article (what does this mean?). If published, this will include your full peer review and any attached files.

Reviewer #1: No

Reviewer #2: No

Reviewer #3: No
---

## [Decision Letter · Decision Letter 1]

6 Jan 2021

Dear Post,

Thank you very much for submitting your manuscript " Infection Manager System (IMS) as a new hemocytometry-based bacteremia detection tool: a diagnostic accuracy study in a malaria-endemic area of Burkina Faso " for consideration at PLOS Neglected Tropical Diseases. As with all papers reviewed by the journal, your manuscript was reviewed by members of the editorial board and by several independent reviewers. The reviewers appreciated the attention to an important topic. Based on the reviews, we are likely to accept this manuscript for publication, providing that you modify the manuscript according to the review recommendations. 

Sincerely,

Yoke Fun Chan, PhD

Associate Editor

Jeanne Salje

Deputy Editor

Reviewer's Responses to Questions

**Key Review Criteria Required for Acceptance?**

**Methods**

-Are the objectives of the study clearly articulated with a clear testable hypothesis stated?

-Is the study design appropriate to address the stated objectives?

-Is the population clearly described and appropriate for the hypothesis being tested?

-Is the sample size sufficient to ensure adequate power to address the hypothesis being tested?

-Were correct statistical analysis used to support conclusions?

-Are there concerns about ethical or regulatory requirements being met?

Reviewer #2: The objectives are clearly stated and tested, with study designs appropriate to address the objectives.

The studied population was clearly described and suitable for the study.

The sample size was large enough to test the stated hypotheses. 

I have suggested a few more statistical analyses to improve the manuscript and support some stands better.

Ethical & regulatory requirements were met.

Reviewer #3: (No Response)

**Results**

-Does the analysis presented match the analysis plan?

-Are the results clearly and completely presented?

-Are the figures (Tables, Images) of sufficient quality for clarity?

Reviewer #2: The text may require further editing to present the messages more clearly. There are a few issues in tables and images that I have stated, which require improvement.

Reviewer #3: (No Response)

**Conclusions**

-Are the conclusions supported by the data presented?

-Are the limitations of analysis clearly described?

-Do the authors discuss how these data can be helpful to advance our understanding of the topic under study?

-Is public health relevance addressed?

Reviewer #2: Appropriate conclusions were drawn.

Limitations are elaborated more clearly in the second version, which help readers to understand how the findings can contribute to the field of clinical hematology, given that a few of the limitations stated are solved. 

The public health relevance is adequately addressed.

Reviewer #3: (No Response)

**Editorial and Data Presentation Modifications?**

Reviewer #2: Minor Revision

The authors had addressed most of the questions and issues raised in my previous reviewer’s report. 

A few minor notes for further improvement:

1. Rules of using brackets: Do not use the same style of brackets when doing double-bracketing. Just like mathematics, there is such rule in English too, whose sequence of use is usually like this: {([ ])}. Please modify those double brackets in the tables accordingly. 

2. Regarding the matter of not showing dataset of viral infections in plots of Figure 3 but only mentioned in text: Honestly, I don't like the idea of mentioning but not showing. However, if this is insisted and agreed upon by the editor, it is fine, but I strongly suggest you to remodel your sentences in the manuscript, as readers will definitely expect the information (including the viral infections) to be available in the figures in its current form, which presents the 3 sets of data on the same ground. My suggestion:

Line 271: “….Among the different infections, median CRP levels of bacterial infections (88 mg/L, IQR 40 – 129; n = ?) were higher than those obtained from malaria cases. Interestingly, samples of viral infections recorded even lower CRP values (9 mg/L, IQR 3-26; n =?).”

And similar style editing for the PCT description. If the viral infection dataset cannot be shown on the plots, the information should not be presented in the same sentence as the bacterial and malaria infections with the same tone of importance. In addition, the mentioned stats in the text should be accompanied by the n number as they seemed to be different from those datasets presented in the graphs, and there are 3 groups of bacterial infections in the graphs, which readers would not be able to know which group the mentioned stats in the text were based on. 

For the plots, you have datasets good enough to run statistical test for a more meaningful comparison, which is your objective in the manuscript. In the plots, please include a dotted straight line (parallel with the X-axis) to show the cut-off values of CRP and PCT that you used in respective graphs, so that readers can appreciate how different the values are between the normal and diseased states. Please run a comparison test to make the claim in the text (line 275-278) meaningful. Run a normality test on the datasets. If the datasets are normally distributed, do a One-way ANOVA with Tukey’s test to cross-compare these data. If the datasets are not normally distributed, perform a Kruskal Wallis stat with Dunn’s test to cross-compare these data. No matter how widespread and overlapped the datasets of these groups are, one needs a statistical analysis to verify if there is any significant difference across the groups. Besides, your plots for PCT in Fig. 3D showed that the medians are quite different. Hence, a proper statistical analysis is really needed here. 

Also in your plots and tables, I noticed that you used different concentration units. For example, in the graph, CRP is presented in mg/dL but in the table, it is in mg/L. Likewise for PCT, µg/L is used in the plot but ng/ml is used in table (although in this case, both units mean the same amount). Please be consistent throughout the manuscript (in text, table and figures). 

3. Authors had the tendency to start a paragraph/ sentence with the words “Figure/ Table”. These look like a repeat of descriptions for the figure/ table legends. Please rephrase them to improve the manuscript. For example, line 315 can be modified like this:

“….We also investigated the impact of malaria infection on the diagnostic accuracy of these methods (Figure 4). For patients without malaria (parasites not detected from peripheral blood via light microscopy), IMS and CRP had comparable accuracy, whereas in patients with malaria, IMS was more accurate than CRP. Of note, PCT was the least accurate in both settings…..”.

Please do not use the term “malaria parasitemia” as used in line 316. If the authors were trying to describe this as “patients harboring parasites detectable via microscopy”, kindly consider the above suggestion of rephrasing. In addition, please change other parts with similar style of presentation accordingly (Line 244, 258, 295). 

4. There are some issues of spacing and usage of punctuation marks throughout the manuscript (e.g. line 40, 73, 81, 94 to name a few). Please correct them.

5. Sentence at line 103: “…..which IS used to create……”

6. Line 242: please remove a redundant set of “with and”

7. Line 406: please remove the redundant “simultaneously”

8. Also at Line 406: “…..decision making at costs similar to that of CBC alone.”

9. Line 413: What is “Alinea”? Is it a typo? Please edit the sentence.

Reviewer #3: (No Response)

**Summary and General Comments**

Reviewer #2: This work is definitely of importance to be published. However, it needs further improvement at statistical analyses, style of data presentation and English to be deemed ready for publication. Nevertheless, at its current state, these minor revisions shouldn't take too long for the manuscript to be accepted for publication.

Reviewer #3: (No Response)

PLOS authors have the option to publish the peer review history of their article (what does this mean?). If published, this will include your full peer review and any attached files.

Reviewer #2: No

Reviewer #3: No
---

## [Editor Report · Decision Letter 2]

30 Jan 2021

Dear Post,

We are pleased to inform you that your manuscript ' Infection Manager System (IMS) as a new hemocytometry-based bacteremia detection tool: a diagnostic accuracy study in a malaria-endemic area of Burkina Faso ' has been provisionally accepted for publication in PLOS Neglected Tropical Diseases.

Best regards,

Jeanne Salje

Deputy Editor

Jeanne Salje

Deputy Editor

---

## [Editor Report · Acceptance letter]

18 Feb 2021

Dear Post,

We are delighted to inform you that your manuscript, " Infection Manager System (IMS) as a new hemocytometry-based bacteremia detection tool: a diagnostic accuracy study in a malaria-endemic area of Burkina Faso ," has been formally accepted for publication in PLOS Neglected Tropical Diseases.

Best regards,

Shaden Kamhawi

co-Editor-in-Chief

Paul Brindley

co-Editor-in-Chief
